# Genome-Based Classification of *Pedobacter albus* sp. nov. and *Pedobacter flavus* sp. nov. Isolated from Soil

**Nhan Le Thi Tuyet and Jaisoo Kim ***

Department of Life Science, College of Natural Sciences, Kyonggi University, Suwon 16227, Republic of Korea; nhanle@kyonggi.ac.kr
* Correspondence: jkimtamu@kgu.ac.kr; Tel.: +82-31-249-9648

**Abstract:** Two rod-shaped, non-spore-forming, Gram-negative bacteria, strain KR3-3[T] isolated from fresh soil in Korea and strain VNH31[T] obtained from soil samples from motorbike repair workshop floors in Vietnam, were identified. Phylogenetic analysis utilizing 16S rRNA gene sequences revealed their affiliation with the family Sphingobacteriaceae and their relation to the genus *Pedobacter*, exhibiting 16S rRNA gene sequence similarities lower than 98.00% with all known species within the genus *Pedobacter*. Growth of VNH31[T] and KR3-3[T] was impeded by NaCl concentrations exceeding >0.5% and 1.5%, respectively, while they both thrived optimally at temperatures ranging between 25 and 30 °C. Notably, neither strain reduced nitrate to nitrite nor produced indole. Negative results were observed for the acidification of D-glucose and hydrolysis of urea, gelatin, casein, and starch. VNH31[T] exhibited growth on β-galactosidase, sodium acetate, L-serine, and L-proline, whereas KR 3-3[T] demonstrated growth on D-glucose, D-mannose, D-maltose, N-acetyl-glucosamine, sucrose, sodium acetate, L-serine, 4-Hydroxybenzoic acid, and L-proline. Core genome-based phylogenetic analysis revealed that the two isolates formed distinct clusters within the genus *Pedobacter*. The DNA G+C contents of KR3-3[T] and VNH31[T] were determined to be 44.12 mol% and 32.96 mol%, respectively. The average nucleotide identity and in silico DNA-DNA hybridization relatedness values (67.19–74.19% and 17.6–23.6%, respectively) between the *Pedobacter* isolates and the closely related type strains fell below the threshold values utilized for species delineation. Following comprehensive genomic, chemotaxonomic, phenotypic, and phylogenetic analyses, the isolated strains are proposed as two novel species within the genus *Pedobacter*, named *Pedobacter albus* sp. nov. (type strain KR3-3[T] = KACC 23486[T] = NBRC 116682[T]) and *Pedobacter flavus* sp. nov. (type strain VNH31[T] = KACC 23297[T] = CCTCC AB 2023109[T]).

**Keywords:** *Pedobacter*; new species; soil; taxonomy; phylogeny; *Sphingobacteriaceae*; *Sphingobacteria*

## 1. Introduction

The genus *Pedobacter* is a genus with increasing numbers of newly described species and is currently accommodating different species assigned to the genus *Pedobacter*, of which 101 are listed with validly published and correct names (https://lpsn.dsmz.de/genus/pedobacter; accessed on 16 March 2024) [1]. This genus belongs to Gram-negative, rod-shaped bacteria, and members of this genus contain both oxidase-positive and oxidase-negative species, as well as non-motile or motile species that move by gliding. The colonies form either pale white or yellow-to-pink pigmented colors on R2A agar [2]. The species of *Pedobacter* can be found in many different habitats such as soil [3–5], water [5–7], sludge [8], nitrifying inoculum [9], fragmentary rock [10], deep sea sediment [11], plant roots [12], and extreme places [13,14]. The wide distribution of *Pedobacter* members in many different habitats might indicate their resistance to environmental factors by developing nutritional versatility, which makes them adaptable in different ecosystems [14].

This study aims to characterize two isolates, namely KR3-3[T] and VNH31[T], isolated from different soils in Korea and Vietnam and delineate their taxonomic positions to determine whether they represent novel species of the genus *Pedobacter*.

## 2. Materials and Methods

### 2.1. Isolation and Cultivation

This study presents a taxonomic investigation of strains KR3-3[T] and VNH31[T], which were isolated from soil in Korea and Vietnam. Strain KR3-3[T] originated from fresh soil collected at Cheongggye-dong, Uiwang-si, Gyeonggi-do, Republic of Korea (GPS coordinates: 37°24′52.6″ N, 127°02′29.4″ E) in June 2023. Strain VNH31[T] was isolated from a soil sample taken from a motorbike repair workshop floor in Phu Vang, Thua Thien Hue, Vietnam (17°04′16.3″ N, 106°59′28.4″ E) in April 2022. Isolation involved placing a debris-free sieved soil sample (3 g) at the bottom of a six-transwell plate (Corning Inc., New York, NY, USA), adding 3 mL of 50% R2A, and subsequently inoculating 100 µL of the soil suspension into the insert. The cultures were then incubated at 25 °C in a shaking incubator at 130 rpm for 2 weeks. Following incubation, 100 µL of each diluted culture was spread onto R2A agar plates and cultivated at 28 °C for up to 5 days. Individual colonies exhibiting distinct morphologies were selected and streaked on R2A medium at 28 °C until pure colonies were obtained. The resulting pure cultures were preserved at −70 °C with 20% (*v/v*) glycerol until further analysis.

Strain KR3-3[T] was deposited in the Korean Agricultural Culture Collection and the National Institute of Technology and Evaluation (Japan) for sharing with other researchers for systematic experiments [15]. They were given accession numbers KACC 23486[T] and NBRC 00000[T], respectively. Strain VNH31[T] was deposited in the Korean Agricultural Culture Collection and the China Centre for Type Culture Collection under accession numbers KACC 23297[T] and CCTCC AB 2023109[T], respectively. The reference type strains *Pedobacter solisilvae* KCTC 42612[T], *Pedobacter insulae* KACC 14010[T], and *Pedobacter cryotolerans* KACC 19998[T] were obtained from the Korean Collection for Type Cultures (KCTC) and the Korean Agricultural Culture Collection (KACC) for comparative studies.

### 2.2. Gene Sequencing and Phylogenetic Analysis of 16S rRNA

Amplification of the 16S rRNA gene fragment from strains KR3-3[T] and VNH31[T] was carried out using the universal primers 27F (5′-AGAGTTTGATCMTGGCTCAG-3′) and 1492R (5′-TACGGYTACCTTGTTACGACT T-3′), followed by sequencing of the resulting DNA amplicons conducted by Macrogen (Seoul, Republic of Korea). To identify the most closely related type strain of 16S rRNA gene sequences, the 16S rRNA sequencing data of KR3-3[T] and VNH31[T] were compared against the EzBioCloud 16S rRNA gene sequence database (www.ezbiocloud.net/eztaxon, accessed on 7 January 2024) [16]. The 16S rRNA gene sequences of the reference strains were retrieved from their entries in the NCBI GenBank database (www.ncbi.nlm.nih.gov/, accessed on 7 January 2024). Subsequently, the 16S rRNA gene sequences were aligned using the SILVA aligner [17]. Phylogenetic analysis was conducted using Mega 11 software, generating three major phylogenetic trees employing the maximum likelihood (ML), neighbor-joining (NJ), and minimum evolution (ME) methods. Bootstrap analyses were performed with 1000 resamplings to determine the bootstrap values, and the evolutionary distances were calculated using the Kimura two-parameter method [18].

### 2.3. Genome Sequence Analysis

A TruSeq DNA PCR-Free kit (Novogene, Beijing, China) was utilized for analysis of the draft genome sequences of strains VNH31[T] and KR3-3[T]. The draft genome sequences were processed using the Illumina HiSeq X platform and assembled by employing the SPAdes version 3.13.0 de novo assembler at Macrogen Co., Ltd. (Seoul, Republic of Korea). The average nucleotide identity (ANI) values and digital DNA-DNA hybridization (dDDH) values between KR3-3[T] and VNH31[T], as well as between these strains and the type strains of the *Pedobacter* genus, were calculated using EzBioCloud [19] and the GGDC web server [20], respectively. Annotation of the genome sequences of KR3-3[T] and VNH31[T] was conducted using the Rapid Annotation with Subsystem Technology (RAST) server version 2.0 [21]. The antiSMASH server version 7.1 was used to identify the biosynthetic gene clusters (BGCs)

for various secondary metabolites [22]. Cluster of orthologous group (COG) analyses were performed to classify genes based on their functions, utilizing the Kyoto Encyclopedia of Genes and Genomes (KEGG) database [23]. For the analysis of evolutionary divergence based on whole-genome sequences, phylogenetic trees were constructed in silico for the two isolates and the closely related type strains within the *Pedobacter* genus obtained from the NCBI, employing the UBCG pipeline [24] with a concatenated alignment of 92 core genes.

### 2.4. Physiology and Chemoaxonomy

The biochemical and physiological characteristics of strains KR3-3[T] and VNH31[T] were assessed using a battery of tests pertinent to Gram-negative, rod-shaped bacteria. Following cultivation on R2A agar plates at 30 °C for 3 days, the colony morphologies of both strains were observed. The cell morphologies were examined under both light microscopy (BX50; Olympus, Tokyo, Japan) and transmission electron microscopy (Bio-TEM H-7650; Hitachi, Tokyo, Japan) utilizing cells cultured for 2 days at 30 °C on R2A agar (MBcell, Seoul, Republic of Korea). Gram staining was conducted using Hucker's method [25], while endospore formation was assessed by staining with malachite green following the procedure outlined by Schaeffer and Fulton [26]. Screening for the production of flexirubin-type pigment was performed using a 20% KOH test [27].

The oxidase activity was determined using 1% ($w/v$) tetramethyl-p-phenylenediamine, and the catalase activity was assessed based on the formation of bubbles upon mixing a pellet of fresh cell culture with a drop of 3% ($v/v$) hydrogen peroxide ($H_2O_2$). The motility of the cells was examined in R2A medium containing 0.4% agar after incubation for 72–96 h at 30 °C. Additionally, various conventional tube and plate tests were conducted, including starch (Sigma-Aldrich, St. Louis, MO, USA) and casein (1% skimmed milk; MB Cell) hydrolysis, urea hydrolysis, Tween 80 (1%; Sigma-Aldrich) hydrolysis, and DNA testing (CM321, Oxoid).

The isolates were cultured for 5 days at 28 °C on different agar media (R2A agar, tryptone soya agar (TSA; MB Cell), Luria Bertani agar (LB; MB Cell), nutrient agar (NA; MB Cell), and MacConkey agar (MB Cell)) to determine their optimal growth media. Temperature-dependent growth was assessed on R2A agar at various temperatures ranging from 4 to 45 °C. The pH levels and salinity-dependent growth were evaluated in R2A broth incubated at 30 °C at 130 rpm, with the wavelength at an optical density of 600 nm (OD600) measured after 5 days using a spectrophotometer (CARY 300; Varian, Palo Alto, CA, USA). The pH range was adjusted from pH 3 to 11 (in 0.5 pH unit intervals) using specific buffer solutions, while salinity-dependent growth was tested with NaCl concentrations ranging from 0 to 7% ($w/v$) in 0.5% intervals. The specific buffer solutions were citrate/$NaH_2PO_4$ buffer (for pH 3.0–5.5), Sorensen's phosphate buffer (for pH 6.5–8), Tris buffer (for pH 8.5–9), carbonate buffer (for pH 9.5–10.0), and 5 M NaOH (for pH 10.5–11).

Additional physiological characterization of the strains was conducted by employing API 20NE and API ID 32 GN strips (bioMérieux, Marcy-l'Étoile, France) to assess fundamental chemical tests and carbon source utilization. The enzyme activities of the isolates and sugar acidification were further examined using API ZYM test strips (bioMérieux). These analyses were carried out following the guidelines provided by the manufacturers for the commercial kits API 20NE, API ID 32 GN, and API ZYM.

To analyze the quinones and polar lipids, cells of KR3-3[T] and VNH31[T] were cultured on TSB agar at 30 °C for 72 h. Quinones were extracted from 100 mg of freeze-dried cells following the method outlined by Minnikin et al. (1984) [28] and analyzed using an Agilent 1260 infinity HPLC system. Polar lipids were extracted and analyzed through thin-layer chromatography (TLC), following the comprehensive procedure detailed by Minnikin et al. (1984) [24]. To profile cellular fatty acids, strains KR3-3[T] and VNH31[T], along with the reference strains of closely related species, were grown under identical conditions on TSB agar (Oxoid) at 30 °C for 72 h, and the cells were then harvested. Fatty acids were extracted using the Sherlock Microbial Identification (MIDI) system protocol and analyzed

using an Agilent 6890 N gas chromatography system in conjunction with the Microbial Identification software package and the Sherlock system MIDI 6.3, utilizing the Sherlock Aerobic Bacterial Database (TSBA6.21) [29].

## 3. Results and Discussion

### 3.1. Molecular Sequencing and Genome Characteristics

A preliminary examination of the 16S rRNA gene indicated that KR3-3[T] (OR553950) comprised 1446 nucleotides, while VNH31[T] (OQ683847) encompassed 1468 nucleotides. Both strains were classified within the family Sphingobacteriaceae, falling under the Sphingobacteriia subdivision. Upon comparison, the top matches for all sequences were with putative isolates belonging to the genus *Pedobacter*. Specifically, KR3-3[T] demonstrated the highest similarity to *Pedobacter solisilvae* (98.06%), whereas VNH31[T] exhibited the closest relation to *Pedobacter antarcticus* (96.12%). Similarities with other *Pedobacter* species were from 96.05% and below. These values notably fell below the suggested threshold of 98.7% similarity used for defining bacterial species [30], indicating that the isolates represented novel species.

Phylogenetic analysis was conducted using Mega 11 software. The evolutionary lineage was deduced by employing the maximum likelihood (ML) method, as depicted in Figure S1. The ML tree's structure was corroborated by the ME (Figure S2) and NJ trees (Figure S3). In the ML and NJ trees, strain KR3-3[T] appeared to be distinct from other closely related *Pedobacter* species. *Pedobacter solisilvae* emerged as its nearest phylogenetic relative, supported by a 100% bootstrap value. Meanwhile, strain VNH31[T] was closest to the type strains of *Pedobacter antarcticus*, *Pedobacter psychroterrae*, *Pedobacter ginsengisoli*, *Pedobacter seoulensis*, and *Pedobacter schmidteae*, with bootstrap support of 52%.

However, whole-genome phylogenetic analysis revealed that VNH31[T] and KR3-3[T] formed distinct clusters which occupied distant positions in the phylogenetic tree and exhibited different close strains compared with those in the 16S phylogenetic tree. VNH31[T] showed its closest relationships with *P. glucosidilyticus*, *P. aquae*, *P. cryophilus*, *P. arcticus*, *P. segetis*, *P. indicus*, *P. mongoliensis*, and *P. xinjiangensis*, whereas KR3-3[T] was most closely related to *P. insulae*, *P. poleris*, *P. cryotolerans*, *P. frigiditerrae*, *P. planticolens*, and *P. boryungensis* (Figure 1). Notably, the *P. solisilvae* genome was not included in this comparison. Current trends suggest that 16S rRNA gene-based phylogeny and similarity may not be reliable for determining the phylogenetically closest relatives of new lineages, and thus a genome-wide comparison could provide more reliable insights based on key criteria. For comparative analysis in terms of physiology, biochemical characteristics, quinones, and fatty acids, reference organisms such as *P. solisilvae* KCTC 42612[T], *P. insulae* KACC 14010[T], *P. cryotolerans* KACC 19998[T], *P. cryophilus* KACC 19999[T], and *P. mongoliensis* KCTC 52859[T] were selected for comparison based on the availability of their whole-genome data, with the exception of *P. solisilvae* KCTC 42612[T] due to no WGS being available.

The genome of KR3-3[T] had 6221 protein-coding genes, with 5780 (92.9%) assigned to COG categories, whereas for VNH31[T], the corresponding numbers were 2270 and 2150 (94.7%), respectively. The distribution of genes into COG categories showed that 4.1% (239 genes) and 6% (130 genes) in KR3-3[T] and VNH31[T], respectively, were related to general function prediction. Additionally, amino acid transport and metabolism accounted for 7.2% (421 genes) and 7.1% (153 genes), inorganic ion transport and metabolism accounted for 5.2% (303 genes) and 4.0% (86 genes), and energy production and conversion accounted for 5.4% (315 genes) and 4.3% (94 genes) in KR3-3[T] and VNH31[T], with 29.9% and 33.8% of the genes having unknown functions within the COGs (Figure S6), respectively. Pairwise comparisons of the ANI and dDDH values between strains KR3-3[T] and VNH31[T] revealed percentages of 69.86% and 20.00%, respectively. When compared with other type strains of the genus *Pedobacter*, the ANI values ranged from 67.19% to 74.19%, and the dDDH values ranged from 17.6% to 23.6% (Table S1). As suggested, an ANI of <93% and a dDDH identity of 70% may indicate intragenus but interspecies distinctions, indicating

that the two isolated strains represent novel *Pedobacter* species distinct from known species of this genus.

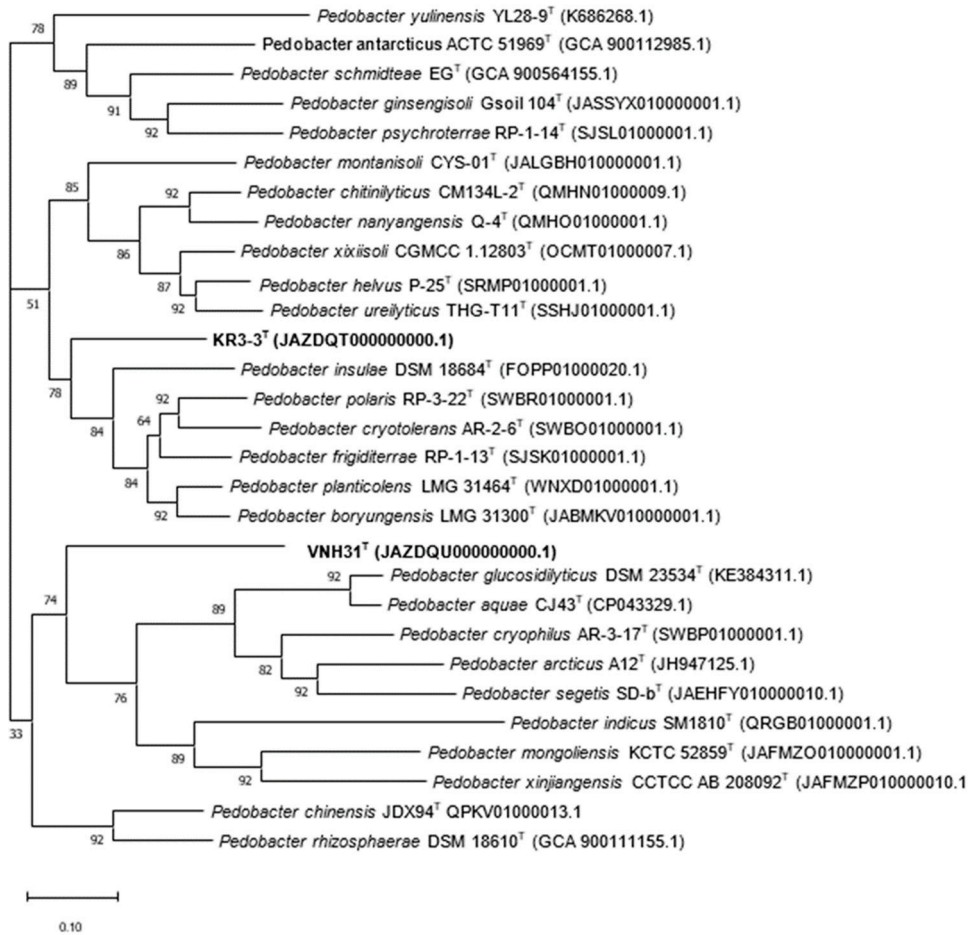

**Figure 1.** The UBCG was utilized to reconstruct the core genome-based phylogenetic tree, incorporating the concatenated alignment of 92 core genes. This tree illustrates the positioning of the KR3-3$^T$ and VNH31$^T$ strains alongside closely related *Pedobacter*-type strains with validly published names, with their corresponding GenBank accession numbers provided in parentheses. The scale bar represents 0.1 nucleotide substitutions per site.

The size of strain VNH31$^T$'s complete genome was 2,435,962 bp, with an N50 value of 901,739 kb, genome coverage of 25.03×, and a DNA G+C content of 32.96 mol%. In contrast, strain KR3-3$^T$ had a complete genome size of 4,576,970 bp, an N50 value of 947,644 kb, genome coverage of 13.32×, and a DNA G+C content of 44.12 mol%. RAST analysis identified the genome of strain VNH31T as having 2306 coding sequences, 39 RNAs, and 204 subsystems. Nevertheless, only 24% of this genome was annotated, and the other 76% was not assigned to the RAST subsystems. The most represented subsystem features were amino acids and their derivatives (143), protein metabolism (122), cofactors, vitamins, prosthetic groups (108), and carbohydrates (67) (Figure S4A). These values for strain KR3-3T were 4063 coding sequences, 41 RNAs, and 230 subsystems. Of this, 18% of this genome was annotated, and the other 82% was not assigned to the RAST subsystems. The most represented subsystem features were amino acids and derivatives (180), protein metabolism (127), cofactors, vitamins, prosthetic groups (124), and carbohydrates (101) (Figure S4B). Screening the genes of the secondary metabolites showed two strains having different gene clusters. VNH31T had a gene cluster belonging to the terpene biosynthesis-related clusters (838,610–859,446 nucleotides), being 28% similar to the known BGC (BGC0000650), which is a gene cluster comprising carotenoids biosynthetic carotenoids. Nevertheless, the low similarity of the predicted gene cluster may represent the production of new metabolites

(Figure S5A). KR3-3T displayed orphan biosynthetic gene clusters (BGCs), which could not identify the known homologous gene cluster (Figure S5B).

### 3.2. Phenotypic Characteristics

On the R2A agar medium, strains VNH31$^T$ and KR3-3$^T$ formed circular, convex, glistening, regular margins and smooth surfaces within 48 h at 28–30 °C. The colonies' colors were white for strain KR3-3$^T$ and yellow for strain VNH31$^T$ (Figure S7). One-week-old colonies of these bacteria were 1–5 mm in diameter for KR3-3$^T$ and 2–6 mm in diameter for VNH31$^T$. The cells were non-spore-forming, non-motile, rod-shaped, stained Gram-negative, and facultatively anaerobic. The cells of strain VNH31$^T$ were slightly larger (0.3–0.9 × 0.8–4.0 μm) than those of strain KR3-3$^T$ (0.3–0.7 × 0.8–3.0 μm) (Figure S8). No flexirubin-type pigments were revealed in the cells of strain VNH31$^T$. KR3-3$^T$ grew well on the R2A, TSA, NA, and LB agar but did not grow on the MacConkey agar. Good growth occurred on the R2A agar, weak growth occurred on the TSA and NA agar for VNH31$^T$, and it failed to grow on the LB and MacConkey agar. Strain KR3-3$^T$ differed from strain VNH31$^T$ in its ability to hydrolyze aesculin and Tween 80. All of the novel strains were able to hydrolyze DNA but not starch or casein. Both strains displayed catalase-positive characteristics, and KR3-3$^T$ was positive with oxidase, whereas a weak oxidase-positive reaction was observed for strain VNH31$^T$. Strain VNH31$^T$ grew in the pH range of 6.5–9.0 with optimum growth at pH 8.0–8.5, whereas strain KR3-3$^T$ grew at 6.0–8.5, with a larger rank of optimum growth at pH 6.5–8.0. The temperature range for growth was 15–37 °C, with optimum growth at 25–30 °C for strain VNH31$^T$, while the values for KR3-3$^T$ were 10–35 °C and 25–30 °C, respectively. Growth of strain KR3-3$^T$ was completely inhibited at NaCl concentrations above 1.5% (*w/v*), but VNH31$^T$ was inhibited at low concentrations above 0.5% (*w/v*). The phenotypic features of the two isolates compared to their closely related type species are presented in detail in Table 1 and Table S2. None of the strains reduced nitrates to nitrites or demonstrated indole production or glucose fermentation. Some differential properties of the two isolated strains in the enzymic activities (API ZYM) consisted of alpha-galactosidase, beta-galactosidase, beta-glucosidase, and alpha fucosidase, which were detected in KR3-3$^T$ but not in VNH31$^T$, whereas alpha chymotrypsin was detected in VNH31$^T$ but not in KR3-3$^T$. Similar to carbon source utilization, KR3-3$^T$ was able to use D-mannose, D-maltose, N-acetyl-glucosamine, sucrose, D-glucose, salicin, D-melibiose, and 4-hydroxybenzoic acid, but VNH31$^T$ was not. When comparing KR3-3$^T$ and *Pedobacter solisilvae*, KCTC 42612$^T$ showed a significant difference in its phenotypic properties in the use of L-serine and 4-hydroxybenzoic acid and the enzymic activities of trypsin and alpha chymotrypsin. These characteristics partly show that they are separate strains.

Similar to members of the *Pedobacter* genus, the new isolates contained menaquinone-7 (MK-7) as the predominant quinone. The main polar lipids included phosphatidylethanolamine (PE), unidentified glycolipids (GLs), unidentified amino lipids (ALs), unidentified phospholipids (PLs), and unidentified polar lipids (Ls) (Figure S9). The major components of the cellular fatty acids of the isolates VNH31$^T$ and KR3-3$^T$ were iso C$_{15:0}$ (which constituted 28.6% and 27.36% of the total fatty acids, respectively) and iso C$_{17:0}$ 3OH (10.69% and 11.53%, respectively). Significant amounts of C$_{16:0}$ (2.48% and 3.49%, respectively), anteiso C$_{15:0}$ (4.08% and 2.913%, respectively) and iso C$_{15:0}$ 3OH (3.31% and 2.51%, respectively) were also detected in the cells of the isolates. Aside from this, there was a slight difference between strains VNH31$^T$ and KR3-3$^T$ in the presence of C$_{18:0}$ (3.12% and 0.31%, respectively). The fatty acid compositions of the isolates were found to be similar to those of their closely related type species. However, it was also found that there were differences in the proportion of major fatty acids and the presence or absence of certain minor fatty acids. Accordingly, these findings indicate that the isolates had differences in their closely related type species as well (Table 2).

**Table 1.** Phenotypic characteristics differentiating the novel species VNH31[T] and KR3-3[T] from other closely related species. A detailed description of results from the API ZYM, API 20NE, and API 32GN test kits are provided in Table S2. Strains: 1 = VNH31[T]; 2 = KR3-3[T]; 3 = *Pedobacter solisilvae* KCTC 42612[T]; 4 = *Pedobacter insulae* KACC 14010[T]; 5 = *Pedobacter cryotolerans* KACC 19998[T]; (Data were from the present study, except the G+C content of the reference strains, which were retrieved from the literature in parentheses.) 6 = *Pedobacter cryophilus* KACC 19999[T]; 7 = *Pedobacter mongoliensis* KCTC 52859[T]. + = positive; − = negative; +/− = weak positive or ambiguous; nd = no data.

| Characteristics | 1 | 2 | 3 | 4 | 5 | 6 | 7 |
|---|---|---|---|---|---|---|---|
| Colony color on R2A | Yellow | White | White | Light yellow | Yellow | Rose red | Light pink |
| Temperature range for growth (°C) | 15–37 | 10–35 | 15–35 | 10–30 | 4–28 | 4–28 | 4–37 |
| Optimal growth temperature (°C) | 25–30 | 25–30 | 25–30 | 25–28 | 15–20 | 15–20 | 28 |
| pH range for growth | 6.5–9.0 | 6.0–8.5 | 5.5–9.5 | 6.0–10 | 6.0–10.0 | 6.0–10.5 | 6.0–9.0 |
| pH optimum for growth | 8.0–8.5 | 6.5–8.0 | 7.0–7.5 | 6.5–7.5 | 7.0–9.0 | 7.0–9.0 | 7.0 |
| Highest salt tolerance (%, *w/v*) | 0.5 | 1.5 | 1.5 | 1.5 | 1.5 | 1.0 | 1.0 |
| **Hydrolysis of** | | | | | | | |
| Casein | − | − | + | − | − | − | − |
| Starch | − | − | + | + | − | − | − |
| Aesculin | − | + | + | + | + | + | + |
| Tween 80 | − | + | − | + | − | − | − |
| **Enzymatic reaction** | | | | | | | |
| Esterase (C4) | + | + | + | − | − | +/− | + |
| Crystine arylamidase | + | + | + | − | + | + | − |
| Trypsin | − | − | + | − | + | − | − |
| Alpha chymotrypsin | + | − | + | + | + | − | − |
| Alpha galactosidase | − | + | + | − | − | − | − |
| Beta galactosidase | − | + | + | − | − | + | + |
| Beta glucosidase | − | + | + | + | + | − | + |
| Alpha fucosidase | − | + | + | − | − | − | − |
| **Electron donor or carbon source** | | | | | | | |
| L-arabinose | − | − | − | + | + | + | + |
| D-Mannose | − | + | + | + | + | +/− | + |
| D-maltose | − | + | + | + | + | nd | − |
| Potassium gluconate | − | − | − | + | − | − | − |
| L-rhamnose | − | − | − | − | + | + | − |
| N-acetyl-glucosamine | − | + | + | + | + | nd | + |
| Inositol | − | − | − | + | − | nd | − |
| D-saccharose (sucrose) | − | + | + | + | − | − | − |
| D-maltose | − | + | + | + | + | − | − |
| Sodium acetate | + | + | + | − | − | − | − |
| L-alanine | − | − | − | + | − | − | − |
| Glycogen | − | − | − | + | + | − | − |
| L-serine | + | + | − | + | − | − | − |
| D-glucose | − | + | + | + | + | +/− | + |
| Salicin | − | + | + | + | − | − | − |
| D-melibiose | − | + | + | + | − | − | − |
| L-arabinose | +/− | − | − | + | + | +/− | + |
| 3-hydroxybutyric acid | − | − | +/− | + | − | − | − |
| 4-hydroxybenzoic acid | − | + | − | − | − | − | − |
| L-proline | + | + | + | + | − | + | − |
| DNA G+C content (mol%) | 32.96 | 44.12 | 46.1 | 39.4 | 34.8 | 33.9 | 43.4 |

**Table 2.** Detailed cellular fatty acid profiles (as percentage of totals) of strains VNH31[T] and KR3-3[T] and closely related reference strains. Strains: 1 = VNH31[T]; 2 = KR3-3[T]; 3 = *Pedobacter solisilvae* KCTC 42612[T]; 4 = *Pedobacter insulae* KACC 14010[T]; 5 = *Pedobacter cryotolerans* KACC 19998[T]. All data are from present study. Fatty acids that represent <0.1% of total in all strains are not shown. = <0.1% or not detected.

| Fatty Acid | 1 | 2 | 3 | 4 | 5 |
|---|---|---|---|---|---|
| **Saturated** | | | | | |
| $C_{10:0}$ | 0.19 | 0.62 | | | |
| $C_{14:0}$ | 0.88 | 1.21 | 1.3 | 1.02 | 0.94 |
| $C_{16:0}$ | 2.48 | 3.49 | 2.73 | 3.24 | 1.62 |
| $C_{18:0}$ | 3.21 | 0.31 | 0.13 | | |
| **Unsaturated** | | | | | |
| $C_{14:1}\ \omega5c$ | | | | | 0.14 |
| $C_{15:1}\ \omega6c$ | 1.11 | 0.55 | | | 0.83 |
| $C_{16:1}\ \omega5c$ | 0.69 | 1.05 | 0.73 | 2.67 | 2.64 |
| $C_{17:1}\ \omega6c$ | | 0.16 | | | 0.29 |
| $C_{17:1}\ \omega8c$ | | 0.23 | | | 0.28 |
| $C_{18:1}\ \omega5c$ | 0.90 | 0.37 | 0.12 | | 0.62 |
| **Branched chain** | | | | | |
| iso $C_{11:0}$ | 1.70 | 0.37 | | | |
| iso $C_{13:0}$ | 0.54 | 0.66 | 0.61 | | 0.4 |
| iso $C_{14:0}$ | 0.35 | 0.20 | | | |
| iso $C_{15:0}$ | 28.68 | 27.36 | 27.73 | 26.83 | 22.32 |
| iso $C_{16:0}$ | 0.92 | 0.43 | 0.19 | 2.22 | 1.68 |
| iso $C_{17:0}$ | | 1.00 | 0.94 | | 0.31 |
| iso F $C_{15:1}$ | 0.46 | | | | |
| iso H $C_{16:1}$ | 0.81 | 0.66 | | | 1.90 |
| anteiso $C_{11:0}$ | 0.45 | 0.41 | | | |
| anteiso $C_{15:0}$ | 4.08 | 2.29 | 1.13 | 8.59 | 2.65 |
| anteiso $C_{17:0}$ | | 0.11 | 0.1 | 0.33 | 0.59 |
| alcohol-$C_{16:1}\ \omega7c$ | | | | | |
| anteiso-$C_{17:1}\ \omega9c$ | 0.72 | 0.78 | | | 3.89 |
| **Hydroxy** | | | | | |
| $C_{8:0}$ 3OH | 0.39 | | | 0.65 | 0.14 |
| $C_{14:0}$ 2OH | | 0.55 | 0.83 | 0.4 | 0.41 |
| $C_{15:0}$ 2OH | | 0.49 | 0.14 | | 1.46 |
| $C_{15:0}$ 3OH | 1.11 | | | | |
| $C_{16:0}$ 3OH | 0.77 | 3.36 | 3.37 | 1.41 | 0.3 |
| $C_{17:0}$ 2OH | 0.66 | 0.98 | 0.43 | 0.89 | 7.43 |
| $C_{17:0}$ 3OH | | 0.15 | | | |
| iso $C_{14:0}$ 3OH | | 0.15 | | | 0.15 |
| iso $C_{15:0}$ 3OH | 3.31 | 2.51 | 2.87 | 1.42 | 2.18 |
| iso $C_{16:0}$ 3OH | 1.50 | 1.17 | 0.45 | | 2.79 |
| iso $C_{17:0}$ 3OH | 10.69 | 11.53 | 10.64 | 8.08 | 8.39 |
| **Summed features *** | | | | | |
| Summed Feature 1 | 0.85 | 0.21 | 0.44 | | 0.5 |
| Summed Feature 3 | 18.57 | 27.02 | 36.94 | 28.60 | 27.57 |
| Summed Feature 4 | | 1.85 | 0.67 | 0.85 | 0.72 |
| Summed Feature 9 | 14.54 | 5.93 | 4.84 | 2.81 | 6.16 |

* Summed features represent groups of two or three fatty acids that could not be separated using the MIDI system. Summed feature 1 contains $C_{13:0}$ 3-OH and/or $C_{15:1}$ i H; summed feature 3 contains $C_{16:1}\ \omega7c$ and/or $C_{16:1}\ \omega6c$; summed feature 4 contains $C_{17:1}$ iso I and/or $C_{17:1}$ anteiso B; summed feature 9 contains iso-$C_{17:1}\ \omega9c$ and/or 10-methyl $C_{16:0}$.

## 4. Conclusions

The newly identified isolates exhibited several distinctive traits setting them apart from phylogenetically linked species within the genus *Pedobacter* (Table 1). KR3-3[T] demonstrated phenotypic differences from *Pedobacter solisilvae* KCTC 42612[T] concerning the utilization of L-serine and 4-hydroxybenzoic acid as well as enzymatic activities such as trypsin and alpha chymotrypsin. Additionally, distinctions were observed in the pairwise ANI values

and digital DNA–DNA hybridization (dDDH) between the novel isolates and other type strains within the genus *Pedobacter*. To summarize, the novel isolates displayed numerous characteristics that distinguished them from other closely related members of the genus *Pedobacter*. Based on the above presented comprehensive polyphasic data, isolates KR3-3[T] and VNH31[T] are proposed as representatives of two novel species within the genus *Pedobacter*, named *Pedobacter albus* sp. nov. and *Pedobacter flavus* sp. nov., respectively. Further investigations may delve into more detailed genomic studies to elucidate their yet undiscovered functions.

### 4.1. Description of Pedobacter albus sp. nov.

*Pedobacter albus* (al'bus. L. masc. adj. albus, white, relating to the color of the colonies)

KR3-3[T] cells display typical characteristics of Gram-negative, rod-shaped bacteria, exhibiting facultative anaerobic behavior, with cell dimensions in the ranges of approximately 0.8–3.0 μm in length and 0.3–0.7 μm in width under both light microscopy and transmission electron microscopy. When cultured on R2A medium, colonies appear as white, circular, and convex with smooth surfaces and regular margins, achieving a size of 0.5–2.5 mm within a 3 day incubation period. Optimal growth conditions include temperatures ranging from 10 to 35 °C (with an optimum of 25–30 °C) and pH levels between 6.0 and 8.5 (with an optimum pH level of 6.5–8.0), exhibiting maximum salt tolerance at 1.5% NaCl. Positive catalase and oxidase activities were observed. While thriving on R2A, TSA, NA, and LB media, growth was inhibited on MacConkey agar. The species exhibited positive hydrolysis of Tween 80 and aesculin, while negative results were observed for starch, casein, urease, and gelatin hydrolysis, as well as nitrate reduction to nitrite and indole production. Analysis using the API 20NE and API 32GN systems indicated positive results for β-galactosidase activity and assimilation of various substrates such as D-glucose, D-mannose, D-maltose, N-acetyl-glucosamine, sucrose, sodium acetate, L-serine, salicin, D-melibiose, 4-hydroxybenzoic acid, and L-proline, while negative results were obtained for L-arabinose, D-mannitol, L-alanine, L-arabinose, and 3-hydroxybutyric acid. In the API ZYM system, the positive enzymatic activities included alkaline phosphatase, esterase (C4), esterase lipase (C8), leucine arylamidase, valine arylamidase, cystine arylamidase, acid phosphatase, naphthol-AS-BI-phosphate, α-glucosidase, N-acetyl beta glucosaminidase, and α-fucosidase, while negative results were obtained for lipase (C14), trypsin, α-chymotrypsin, β-glucuronidase, and α-monosidase.

The designated type strain KR3-3[T] (=KACC 23486[T] = NBRC 116682[T]) was isolated from agricultural soil in Cheongggye-dong, Uiwang-si, Gyeonggi-do, Republic of Korea. The genome of this strain spans 4,576,970 bp, with a G+C content of 44.12 mol%. The GenBank accession numbers for the sequences of *Pedobacter albus* strain KR3-3[T] are JAZDQT000000000 (genome) and OR553950 (16S rRNA gene nucleotide sequence).

### 4.2. Description of Pedobacter flavus sp. nov.

*Pedobacter flavus* (fla'vus. L. masc. adj. flavus, golden yellow, referring to the colony color)

The characteristics of VNH31[T] cells include being Gram- negative, rod-shaped, and facultatively anaerobic, with dimensions approximately in the ranges of 0.8–4.0 μm in length and 0.3–0.9 μm in width. When cultured on R2A medium, the colonies exhibited a yellow color, circular shape, convex morphology, regular margins, and smooth surfaces, reaching sizes ranging from 2 to 6 mm within a 3 day period. Optimal growth conditions were observed at temperatures between 15 and 37 °C (with an optimum range of 25–30 °C) and pH levels ranging from 6.0 to 9.0 (with an optimal pH level of 8.0–8.5), with the highest salt tolerance noted at 0.5% NaCl. Positive catalase activity was observed, while oxidase activity was weakly positive. Growth was robust on R2A agar, with weak growth on TSA and NA agar and no growth observed on LB or MacConkey agar. No flexirubin-type pigments were detected. Negative results were obtained for the hydrolysis of Tween 80, aesculin, starch, casein, urease, and gelatin, as well as for nitrate reduction to

nitrite and indole production. According to the API 20NE and API 32GN systems, positive results were obtained for β-galactosidase, sodium acetate, and L-proline, while negative results were observed for D-mannose, D-maltose, N-acetyl-glucosamine, D-glucose, salicin, and D-melibiose. In the API ZYM system, positive enzymatic activities included alkaline phosphatase, esterase (C4), esterase lipase (C8), leucine arylamidase, valine arylamidase, cystine arylamidase, α-chymotrypsin, acid phosphatase, naphthol-AS-BI-phosphate, α-glucosidase, and N-acetyl beta glucosaminidase, while negative results were obtained for lipase (C14), trypsin, α-galactosidase, β-galactosidase, β-glucuronidase, β-glucosidase, α-monosidase, and α-fucosidase.

The designated type strain VNH31$^T$ (=KACC 23297$^T$ = CCTCC AB 2023109$^T$) was isolated from agricultural soil in Phu Vang, Thua Thien Hue, Vietnam. The genome of this strain spans 4,576,970 bp, with a G+C content of 32.96 mol%. The GenBank accession numbers for the sequences of *Pedobacter flavus* strain VNH31$^T$ are JAZDQU000000000 (genome) and OQ683847 (16S rRNA gene nucleotide sequence).

**Supplementary Materials:** The following supporting information can be downloaded at https://www.mdpi.com/article/10.3390/d16050292/s1. Figure S1: The phylogenetic tree was reconstructed with the maximum likelihood method based on 16S rRNA gene sequences of strains KR3-3$^T$ and VNH31$^T$ and type strains of the genus *Pedobacter*. Figure S2: The phylogenetic tree was reconstructed with the minimum evolution method based on 16S rRNA gene sequences of strains KR3-3$^T$ and VNH31$^T$ and type strains of the genus *Pedobacter*. Figure S3: The phylogenetic tree was reconstructed with the neighbor-joining method based on 16S rRNA gene sequences of strains KR3-3 and VNH31 and type strains of the genus *Pedobacter*. Figure S4: Subsystem feature of strains VNH31$^T$ (A) and KR3-3$^T$ (B) revealed by rapid Annotation using Subsystem Technology (RAST) server version 2.0. Figure S5: Secondary metabolism analysis in the genomes of (A) VNH31$^T$ and (B) KR3-3$^T$, using antiSMASH version 7.1 to predict biosynthetic gene clusters. Figure S6: COG functional classification of proteins in KR3-3$^T$ and VNH31$^T$ strain genomes. Figure S5: Colonies of KR3-3$^T$ and VNH31$^T$ were grown on R2A at 30 °C for 72 h. Figure S8: Transmission electron microscopy of VNH31$^T$ and KR3-3$^T$ strains' growth on R2A medium plates for 3 days at 30 °C. Figure S9: Polar lipid profile of strains KR3-3$^T$ and VNH31$^T$. Table S1: ANIb and dDDH values (as percentages) between the genome sequences of the two isolates and other *Pedobacter* species with validly published names. Table S2: Results from API ZYM, API 20NE, and API 32GN testing.

**Author Contributions:** All experiments were conceived, designed, and conducted by N.L.T.T. Both N.L.T.T. and J.K. were involved in interpreting all data, engaging in discussions and editing and approving the final draft of the manuscript. The coordination and supervision of the study were undertaken by J.K. All authors have read and agreed to the published version of the manuscript.

**Funding:** This work was supported by a Kyonggi University Research grant (2021-013).

**Institutional Review Board Statement:** Not applicable.

**Data Availability Statement:** The original contributions presented in the study are included in the article and Supplementary Material, further inquiries can be directed to the corresponding author.

**Conflicts of Interest:** The authors declare no conflicts of interest.

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
