# Peer review of "Genome-Based Classification of Pedobacter albus sp. nov. and Pedobacter flavus sp. nov. Isolated from Soil"

_diversity, doi:10.3390/d16050292_

Round 1

Reviewer 1 Report

Comments and Suggestions for Authors

Manuscript ID: diversity-2976091

The MDPI manuscript ID: diversity-2976091 described Pedobacter albus sp. nov. and Pedobacter flavus sp. nov.. The authors provided sufficient supports for the new taxa. There are some details to be revised.

Q1:  Usually do not advise using keywords that overlap with words used in your research paper. So, the keywords should be adjusted.

Q2:  The introductory segment of the paragraph may benefit from revision to enhance its coherence and flow. I propose rewriting this section accordingly.

Lines 35-37

 “Genus Pedobacter is a genus with increasing numbers of newly described species and is currently accommodating different species assigned to the genus Pedobacter, of which 101 are listed with validly published and correct names”

Q3:

In recent years, a multitude of species within the Pedobacter genus have been documented. Nevertheless, the expansion of genomes has not paralleled this proliferation. Given that this article also encompasses information regarding genomes and analyses relating to COGs (Figure S4), I propose highlighting the potential of this genus to produce compounds relevant to humanity, considering the existence of published studies on the subject.

Suggested reference: https://doi.org/10.3390/microorganisms11102445

Author Response

Comments and Suggestions for Authors

Manuscript ID: diversity-2976091

The MDPI manuscript ID: diversity-2976091 described Pedobacter albus sp. nov. and Pedobacter flavus sp. nov.. The authors provided sufficient supports for the new taxa. There are some details to be revised.

Q1:  Usually do not advise using keywords that overlap with words used in your research paper. So, the keywords should be adjusted.

Answer: changed some keywords as suggested

Q2:  The introductory segment of the paragraph may benefit from revision to enhance its coherence and flow. I propose rewriting this section accordingly.

Line 35-37

 “Genus Pedobacter is a genus with increasing numbers of newly described species and is currently accommodating different species assigned to the genus Pedobacter, of which 101 are listed with validly published and correct names”

Answer: changed as suggested

Q3: In recent years, a multitude of species within the Pedobacter genus have been documented. Nevertheless, the expansion of genomes has not paralleled this proliferation. Given that this article also encompasses information regarding genomes and analyses relating to COGs (Figure S4), I propose highlighting the potential of this genus to produce compounds relevant to humanity, considering the existence of published studies on the subject.

Suggested reference: https://doi.org/10.3390/microorganisms11102445

Answer: revised as suggested through RAST analysis

Reviewer 2 Report

Comments and Suggestions for Authors

I found the presented manuscript very interesting and clearly presented. Several redaction issues should be solved before acceptance:

line 17 - "urea" instead of "urease"

lines 37-38, 67-69, 72, 84, 86, 87, 101, 108 URL-links should be formatted as references

line 130 - "their"

lines 146-147 - Why does TSB cultures  were used to analyze quinones and polar lipids, while other test were performed with R2A cultured cells?

line 226 - anaerobic

line 232 - DNA

Author Response

Comments and Suggestions for Authors

I found the presented manuscript very interesting and clearly presented. Several redaction issues should be solved before acceptance:

line 17 - "urea" instead of "urease"

Answer: changed as suggested

lines 37-38, 67-69, 72, 84, 86, 87, 101, 108 URL-links should be formatted as references

Answer: revised as suggested except several places that would be included usually because of access dates.

line 130 - "their"

Answer: revised as suggested

lines 146-147 - Why does TSB cultures were used to analyze quinones and polar lipids, while other test were performed with R2A cultured cells?

Answer: We used TSB agar for culturing cells to analyze quinone, polar lipids, and fatty acids because TSB is a rich nutrition medium (when compared to R2A) and standard medium for those analyses.

line 226 – anaerobic

Answer: revised as suggested

line 232 – DNA

Answer: revised as suggested